# Maximization of FDM-3D-Objects Gonio-Appearance Effects Using PLA and ABS Filaments and Combining Several Printing Parameters: “A Case Study”

**DOI:** 10.3390/ma12091423

**Published:** 2019-05-01

**Authors:** Bàrbara Micó-Vicent, Esther Perales, Khalil Huraibat, Francisco Miguel Martínez-Verdú, Valentín Viqueira

**Affiliations:** 1Colour and Vision Group, University of Alicante, Ctra. San Vicente del Raspeig s/n, 03690 San Vicente del Raspeig (Alicante), Spain; esther.perales@ua.es (E.P.); khalil.huraibat@ua.es (K.H.); verdu@ua.es (F.M.M.-V.); valentin.viqueira@ua.es (V.V.); 2Departamento de Estadística e Investigación Operativa Aplicadas y Calidad, Universitat Politècnica de València, Campus d’Alcoi. Pl. Ferràndiz y Carbonell s/n, 03801 Alcoy (Alicante), Spain

**Keywords:** FDM, gonio-appearance, factorial design, 3D-printing, multilevel factorial design

## Abstract

In order to consider 3D objects from suitable Fused Deposition Modelling (FDM) printers as prototypes for the automotive sector, this sample must be able to reproduce textural effects (sparkle or graininess) or metallic or gonio-appearance to reinforce the attractive appeal of these materials. This study worked with two different commercial filaments: grey metallic PLA (poly(lactic acid)) and ABS (acrylonitrile-butadiene-styrene copolymer) with diffractive pigments. For both materials, a statistical design of experiments (DoE) was carried out to find the printing parameters effect on the final 3D-objects gonio-appearance. The selected printing parameters were printing speed (2 levels), layer height (2 levels) and sample thickness (3 levels). Twelve smooth square objects were printed from each material. The ABS-diffractive filaments achieved the most significant flop and higher sparkle values than metallic PLA. Graininess was high when working with PLA filaments instead of ABS. Layer height was the most significant parameter to maximize PLA objects’ flop or sparkle effects. The best result was found when printing at 0.1 mm. For the ABS samples, the stronger flop and sparkle effects were achieved with the 50 mm/s printing speed, the 0.1 mm layer height and the lowest thickness level. This study shows the methodology to study the printing parameters effects and interactions to maximize the FDM-3D-objects gonio-appearance.

## 1. Introduction

Additive manufacturing (AM), or 3D printing, has been increasingly adopted in the aerospace, automotive, energy and healthcare industries in recent years. AM 3D printing techniques work by fusing drops, particles or fused filaments to build fused layers with selected shapes and properties. Printing human bones, living 3D implants, turbine components (using superalloys) or automotive engine prototypes are some examples of the unlimited applications of AM [1]. 

New applications are continuously being developed in AM. Working on developing low-cost 3D printers and materials extends its application to schools, homes, libraries and laboratories. The goal of low-cost rapid prototyping is the opportunity for consumers to design and develop functional prototypes. There are huge public libraries with free access to designs and tutorials with 3D-printing advice, etc. 3D printing provides manufacturers with the opportunity to offer customized low-cost products to end users. Increasing attention is paid to this technology, especially in the automotive sector because of the challenge of designing and customizing several car parts [2]. 

The most common 3D printing technology is fused deposition modelling FDM because of the low-cost machine and polymer filaments. Only one biopolymer is commercially available and has already been tested by Fused Deposition Modelling (FDM) technologies and is based on PLA: poly(lactic acid). This material is less frequently used than synthetic counterparts due to the lack of technical performance, particularly poor thermal resistance and mechanical performance. However, the mechanical properties of 3D objects from FDM still need improvement [3]. The main drawback in FDM is the final surface roughness, which depends on the 3D printer’s minimum layer height [4]. Some surface finishes can be used to improve surface roughness, such as acetone treatments with ABS (acrylonitrile-butadiene-styrene copolymer) filaments, or by using transparent coatings [5]. Another important limit of FDM printers is the lack of thermoplastic polymers available for this technology [6]. 

Historically, industrial sectors like automotive or building have used AM as an integral tool in the design process. The automotive industry has become a demanding sector of 3D printing technologies. The process opens up a new world of freedom in design and allows concept cars/aircrafts to be built more quickly and more efficiently than with traditional methods. FDM technologies are widely used for concept models and functional prototypes, but also for end-use parts and manufacturing tools [7]. Printing polymeric samples with special effects and a wide color gamut, for example for door handles and dashboard fascia, confer extra value for end consumers and represent a very good business opportunity. ABS is the most common material used for FDM, with which almost 90% of all prototypes are produced, particularly in clinical and automotive applications [8].

The development of new filaments materials for FDM in order to give new colors, or textural effects as metallic surfaces, is a trending topic in the 3D printing industry. In particular, the use of special effect pigments reinforces the attractive appeal of the materials due to goniochromatic effects, i.e., they present notable color changes under different illumination-viewing conditions (gonio-appearance) [9,10,11,12,13]. In this way, it is possible to distinguish between lightness variations due to metallic pigments and hue and chroma variations due to pearlescent, interference or diffraction pigments. Besides this angular dependence on viewing/illumination direction, special-effect pigments also exhibit a visually complex texture. Depending of the geometric nature of the lighting, you can perceive sparkle, under directional conditions, or graininess, under diffuse conditions, or even both combined in most realistic lighting conditions. The sparkle effect is defined as “the aspect of the appearance of a material that seems to emit or reveal tiny bright points of light that are strikingly brighter than their immediate surround and are made more apparent when a minimum of one of the contributors (observer, specimen, light source) is moved” [14]. Therefore, under direct illumination conditions such as bright sunlight, the flakes in a metallic finish glitter creating a sparkling effect. On the other hand, under diffuse illumination such as a cloudy sky, metallic finishes do not sparkle. Instead, they may create a salt and pepper appearance. This effect may be referred to as graininess or coarseness. In particular, graininess or diffuse coarseness is the perceived contrast of the dark-light irregular pattern, with a scale <100 μm [14]. Both sparkle and graininess depend on the flake size, orientation and distribution [15,16,17]. However, very few filament producers offer filaments with effect pigments, diffractive pigments or metallic particles, to produce sparkle effects [18]. Furthermore, any filament producer can guarantee final textural effects on the produced 3D-printed object. The critical factors in FDM printers have been studied to improve the mechanical properties of the 3D object or reduce printing costs [19,20,21,22]. Vasudevarao et al. [23] proposed an experimental design to determine significant factors and their interactions for optimal surface finish of parts fabricated via the Fused Deposition Modelling process. As Rapid Prototyping is moving towards Rapid Manufacturing there is an increasing stress on obtaining good quality parts. Quality of a prototype includes the surface quality, the mechanical strength and dimensional accuracy among other things. Surface Finish is critical not only for better functionality and look, but also for cost reduction in terms of reduced post-processing of parts (this includes sanding, filing etc.) and overall prototyping time reduction also. However, the effect of 3D printing parameters on the final visual appearance has not yet been analysed. The aim of this work was to control the metallic and interference effect on 3D-printed objects by changing certain printing parameters, such as filament material, sample thickness, printing speed, etc.

## 2. Materials and Methods 

We used the free software platform Blender for the 3D objects designs. To build the 3D samples, an Ultimaker 2+ 3D (Ultimaker B.V., Cambridge, MA, USA) was used. This printer is capable of working with different nozzles sizes: 0.25, 0.4, 0.6 and 0.8 mm. This machine works with a 2.85 mm filament diameter. However, by changing printing parameters it is possible to work with different filament diameters. In addition we used CURA software, designed by Ultimaker (Umetani, Bickel, Matusik, 2015), to change the sample and 3D printing parameters.

For this study, work was done with two commercial filaments from different polymers and with distinct properties. PLA grey and metallic filaments were used (2.85 mm in diameter), obtained from Utimaker (UM9015 PLA Silver Metallic RAL 9006), along with ABS grey with 3% diffractive pigments and a diameter of 1.75 mm, which came from Schlenk Metallic Pigments GmbH (trade name: MAXITHEN® SAN702767/10 Metallic Silver, SCHLENK Metallic Pigments GmbH, Barnsdorfer Hauptstraße 5, Germany) (Figure 1). The last filament was not available on the market before our test. Schlenk Company was interested to know the behavior of the diffractive pigments under work conditions. This work is going to provide critical information to the filament makers on the limits of the final effect of this new filament. 

To make the color and textural characterization, and to measure the gonio-apparent effects on the printed samples, a commercial multi-angle spectrophotometer, BYK-mac, was used. This device provides the CIELAB values under the D65 illuminant at six different measurement geometries. These six illumination-detection geometries are designed by CIE as 45°x:−60°, 45°x:−30°,45°x:−20°, 45°x:0°, 45°x:30° and 45°x:65°, respectively, or regarding the specular direction as 45°:as−15°, 45°:as15°, 45°:as25°, 45°:as45°, 45°:as75° and 45°:as110°. This instrument also measures texture effects such as sparkle and graininess effects. Sparkle is defined as the emission of tiny bright points of light that are strikingly brighter than their immediate surround under directional illumination conditions. Thus, the measurement of the sparkle effect by the BYK-mac was performed by illuminating directionally at three different angles (15°, 45° and 75°), counted clockwise from the sample surface. Three parameters were obtained to characterize sparkle, sparkle intensity (Si), sparkle area (Sa) and sparkle grade (Sg), for the three directional geometries. The total size of the small and bright areas per unit area is called the sparkle area. Sparkle intensity is specified as the intensity of the small bright light spots in relation to the intensity of the surrounding less bright area. Sparkle grade is a combination between sparkle area and sparkle intensity following Equation (1). The graininess is defined as the contrast of the dark-light irregular pattern under diffuse illumination conditions. The BYK-mac instrument (BYK-GARDNER GMBH, Geretsried, Germany) uses an integrating sphere to diffusely illuminate the sample and to provide a unique value, G, to characterize the graininess of a sample. In Figure 2, the measurement condition for both color and texture used by the BYK-mac instrument is shown. Finally, another important and most visually striking property of metallic pigments is the lightness flop and it is defined as the decrease in lightness under directional illumination between two extreme angles of observation (Equation (2)).
(1)$$Sg=\sqrt{S_{i}\cdot S_{a}}-0.8$$
*S_i_*; sparkle intensity, *S_a_*; sparkle area.
(2)$$flop\;=\;\frac{2.69\;{\left(L_{15{}^{\circ}}^{*}-L_{110{}^{\circ}}^{*}\right)}^{1.1}}{{\left(L_{45{}^{\circ}}^{*}\right)}^{0.86}}$$

A statistical design of experiment (DoE) was used to select the parameter values that had to be selected for each sample. F-ratio was used to calculate the P-value to find the significance level of the difference in the measured responses under different experiment conditions. The entire statistical test was carried out at a 95% confidence level. For PLA, the printing parameters of printing temperature were set at 220 °C and bed temperature at 70 °C. Then the following factors and levels were selected for the experiment: printing speed 25–50 mm/s^−1^, layer height 0.1–0.2 mm, sample thickness 0.5, 0.8 and 1.6 mm. To obtain samples with the selected different thicknesses, the nozzle diameter was changed to 0.25 and 0.4 mm. With the ABS filaments, work was done at a melting temperature of 257 °C and a bed temperature of 80 °C. The selected levels for the experiment factors in this case were the same as for PLA, except for printing speed 10–15 mm/s^−1^, which had to be lower because of the filament diameter. It was not possible to obtain good complete samples at a faster printing speed. The infill parameter was 0%, but samples were printed with no gap, using shell thickness to fill the samples. With this method there was no pattern in the infill of any sample, and in all samples there was only a change in the direction of the filament deposition during the sample printing. The orientation of the metallic or diffractive pigments depends on the pigment size and shape and the layer height (Figure 3). A multifactorial DoE 2^2^·3^1^ was selected for each material. 

The sample designs were carried out after taking into account the measurement diameter of the multi-angle spectrophotometer. Cubic samples of 500 × 500 mm and different thicknesses were designed. Samples were printed in a Z orientation to achieve the smoothest surface that the 3D printer could provide (Figure 4). 

The surface sample measurements were taken using white and black cards from Neurtek S.A. (Eibar, Gipuzkoa, Spain). The gonio-appearances of the 3D objects were used as a response to DoE optimization. Thus, three characteristic parameters for metallic pigments were considered for the optimization: sparkle grade, graininess and the lightness flop.

## 3. Results and Discussion

### 3.1. DoE Responses

The gonio-appearance effects of all the 3D-printed objects were measured on white and black backgrounds. The sparkle (Sg), graininess (G) and flop values were used as responses to DoE optimization with the PLA (Table 1) and ABS (Table 2) samples. Differences were found depending on the printing parameter in both filaments. However, it was not possible to know which factors produced the significant differences for each parameter without running a DoE analysis.

### 3.2. PLA

The DoE models for all the PLA results had significant factors and excellent adjustments, over 90% R-squared. Depending on the background, different factors and interactions appeared, which would explain the 3D object’s gonio-appearance. For instance, the sparkle at 45° measured on the white background (Sg45_w) was affected mainly by the layer height and sample thickness interaction, as seen in the P-value, which was lower than the significance level (0.05). The print speed and sample thickness interaction that occurred near the significance level should be taken into account to maximize the sparkle level at 45° (Table 3). The sparkle effect at 45° on the black background (Sg45_b) displayed the same layer height and sample thickness (BC) interaction effect as on the white background. For this reason, to increase the sparkle value at 45°, using the lowest layer height (0.1 mm) and the middle sample thickness (0.8 mm) is recommended (Figure 5). 

Other works find that the spatial orientation has a large impact on Build Time in the FDM process [25], and temperature and time could affect the orientation of the metallic particles across the printed surface. In a specific research, it was found that the final surface quality in FDM printed objects is very sensitive to different printing factors such as layer height and the sample surface angle. For instance, the least roughness was achieved using the lowest layer thickness and the maximum printing angle [26]. Moreover, in metallic coatings, flakes tend to orient themselves parallel to the substrate. The physical mechanism behind this orientation effect is mainly related to the film shrinkage during solvent evaporation, and that effect will be similar due to layer shrinkage by the temperature decrease and the several layer depositions in the printed object. However, flake orientation also depends on the flake diameter [27] and flake concentration or pigment type. Silver dollar metallic flakes orient much better than cornflake metallic flakes, coarse metallic flakes orient much better than fine metallic flakes, and large flakes orient themselves much better than small flakes [28]. However, the color formulation (composition) of the filament is not available in commercial samples. 

Due to the printing directions, the orientation of the pigment flakes could be as it is shown in Figure 5. When the layer height is increased the probability of finding pigment flakes with a good orientation increases, improving the Sg_15 effect. The Sg_15 data are stronger than Sg_45 and Sg_75, and represents the sparkle effect from a good orientation of the pigment flakes, in this way, it achieved the objective of this work, to maximize this effect. The Sg_15 effect was higher on the black background because of the contrast increment, as expected. At the lowest thickness (Tk = 1 or Tk = 2) the orientation of the flakes was improved, and this also improved with the layer height, because it is possible to get higher concentrations of well orientated particles (Figure 6). These behaviors and trends were more difficult to find in Sg_45 or Sg_75 results because in these measurement conditions the sparkle that was measured mainly came from the disorientated flakes. 

As we discussed before, printing speed can affect the preferred orientation of the pigment flakes. The flakes showed better orientation with thinner samples and lower speeds over white or black backgrounds. The lower printing speed allows the orientation of the pigment flakes, and this phenomenon was detected well in the samples with more thickness (Tk = 3). At lower thickness’ (Tk = 1 or Tk = 2) a good orientation was achieved with the decrease of the material content and it was possible to increase the printing speed without a significant decrease of the sparkle values (Figure 7). 

The best adjustment was observed for the 75° sparkle effect on the white background. In this case, almost all the analyzed factors and interactions were significant and the R-squared was over 99% (Table 4). However on the black background, the only significant factors for Sg_75 were sample thickness and speed interaction (AC), and the effect was not the same as on the white background. Under both conditions, the maximum Sg_75 was achieved with the thinnest samples (Tk = 1, or 0.5 mm), as was commented in the Sg_15 analysis. 

The same analytic procedure was followed to analyze the graininess and flop values using black and white backgrounds. Table 5 shows the significant factors or the interactions found for each evaluated response, and the levels that had to be selected to increase the textural effects of the 3D-printed objects. The most significant interaction was BC, and the printing speed (A) was only significant with the Sg75_w value. To increase the sparkle effect, using the 0.1 mm layer height and a low or medium sample thickness (0.5–0.8 mm) is recommended. However, to increase the graininess effect it is better to increase sample thickness (0.8–1.6 mm) and use the 0.2 mm layer height. Sparkle and graininess effects should be affected by the same factor levels. The different layer height influence in the graininess effect could be explained by the surface roughness of the FDM samples, which can confuse the measurement equipment and give increased graininess values due to the filament thickness. In addition, to enhance the metallic effect (flop), it is also better to increase sample thickness to 1.6 mm. This can be explained by the different orientation of the metallic flakes in the sample. By increasing sample thickness, it is possible to obtain better orientation in the metallic particles of the filament, which could positively affect the flop value. Moreover, the shorter the layer height, the easier it is to control the orientation of the metallic flakes toward the same printed layer, and with greater particle disorder in the layer, the flop is lighter.

### 3.3. ABS

In the ABS samples, the Sg45_b value was affected significantly by all the selected factors. The most significant influence was caused by the printing speed and layer height (AB) interaction, and the model achieved an adjustment of more than 99% (Table 6). 

As with the PLA samples, a table was built with a summary of the effects. It was remarkable that at 75°, it was not possible to observe any significant factor for the sparkle effect. In these samples, it proved more difficult to find significant differences in the measurements taken on the white background due to lack of contrast, but it was possible to measure on the black background.

The most significant factor to achieve high flop values for Sg75_b was layer height (0.1mm). The lower the layer height, the higher the flop value on the black or white background (Table 7). This may be due to the diffractive pigment orientations in thinner layers. However, this parameter did not suffice to maintain the diffractive effect, and only in sparkle was it possible to see different colors with the change in direct or diffuse illumination. The textural and final metallic effects were the same in both samples PLA and ABS, as seen in Figure 8. The visual appearance should change significantly with the orientation of the angle of measurement, mostly in the ABS filament with diffractive pigments. However, the “rainbow” effect disappears in working conditions. The reflectance spectrum values in VIS range change mainly in parallel in the PLA samples with metallic pigments, and it is not possible to observe significant differences in the band positions or the spectrum shapes with the change of the orientation of measurement [28]. That mainly means that changes in the L*ab value are due to the changes in the orientation of measurement, but no difference in colors can be observed due to the diffractive pigments, as is represented in the example (Figure 9). The effect that was first observed in the filament was lost in the melting process of 3D printing. The different orientations of layers during the printing process did not allow the diffraction effect to be controlled and maintained as in the raw material. 

To increase the sparkle effect at 45° and 75° it is recommended to use the 0.2 mm layer height. To increase the graininess effect it is better to increase the printing speed (50 mm/s^−1^) and use the 0.1 mm layer height. Due to the filament diameter and the ABS properties it was not possible to build samples with 0.5 mm thickness, and the results are from samples of 0.8, 1.6 and 2.4 mm. Furthermore, the surface uniformity was worse than with the PLA filament from the Ultimaker supplier. These differences could explain the opposite effects of the selected factors in the textural effects.

## 4. Conclusions

The aim of this work was to find significant differences in the textural effect of 3D-printed objects using different filament materials with metallic and diffractive pigments, and to discover the optimal conditions that achieve the maximum sparkle, flop and graininess effects. In addition, a filament producer company Schlenk, was interested to know the behavior of the diffractive pigments under work conditions for the filament prototype that they provided for this work. As pigment manufacturers they are focused on innovation to develop coatings and interesting materials for advanced applications in the automotive sector. With a multivariate factorial DoE, it was possible to define the most significant printing factors for both filaments (ABS prototype and PLA with metallic pigments) that improved the gonio-appearance of the 3D-printed objects. The best conditions to increase the metallic effects of the PLA objects at the 15° observation angle are a low layer height (0.1 mm) and low sample thickness. In addition, it is recommended to use a low printing speed to allow the orientation of the pigment particles. The graininess effect is stronger when layer height is increased (0.2 mm), so working at high layer levels is recommended if increasing the diffuse coarseness of surfaces is important. However, this measurement could be affected by the surface roughness of the FDM samples, and this should be taken into account. For this reason, sparkle and flop parameters are better to analyze the effects of printing factors on FDM 3D objects. To increase the flop parameter (on a black or a white background), increasing sample thickness is recommended. 

With the ABS and diffractive pigments, one undesirable effect is loss of the color change effect. However, the sparkle and flop effects remain and enhance the special effects of the printed samples. To achieve the best flop and sparkle effects with ABS, the optimal condition is a high printing speed. However, layer height has to be selected depending on the desirable effect to be increased, i.e., flop or sparkle. To increase sparkle, it is generally better to work at 0.2 mm, but working at 0.1 mm is better to increase the flop values. The methodology that has been shown in this work (DoE), demonstrated a high efficiency in detecting the best printing parameters in order to increase the special effects of the printed samples, and should be replicated for any new filament material with metallic or diffractive pigments.

## Figures and Tables

**Figure 1 materials-12-01423-f001:**
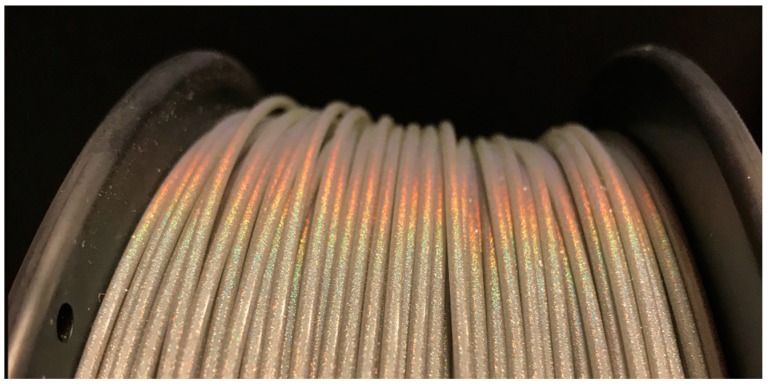
Picture of an ABS (acrylonitrile-butadiene-styrene copolymer) filament (MAXITHEN® SAN702767/10 Metallic Silver).

**Figure 2 materials-12-01423-f002:**
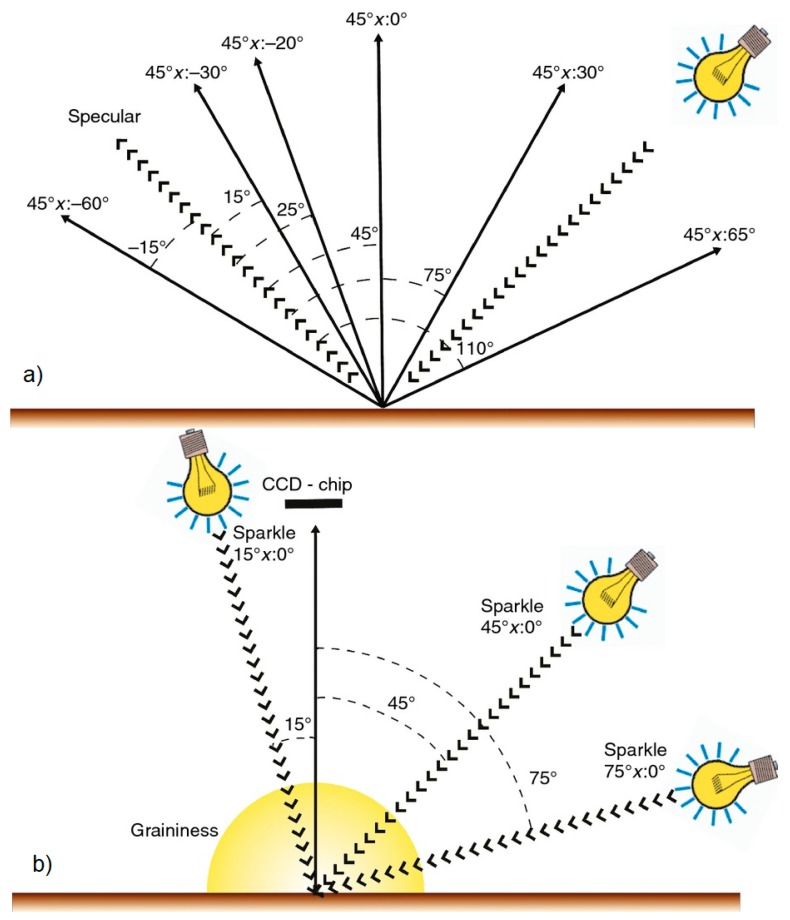
(**a**) Schematic representation of the illumination/measurement geometries used by the BYK-mac device for color characterization. (**b**) Schematic representation of the illumination/measurement geometries used with the BYK-mac device for sparkle and graininess characterization [24].

**Figure 3 materials-12-01423-f003:**
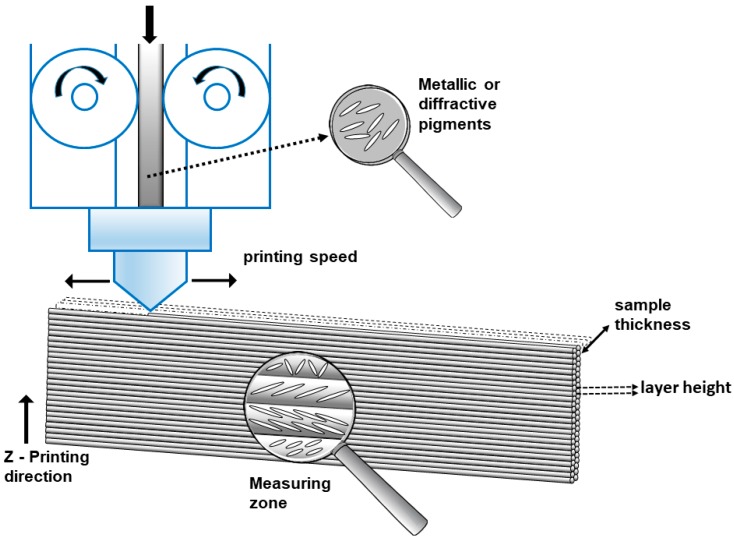
Printing factors schematic with the controlled and changed parameters and the possible disposition of the metallic/diffractive pigments across the printed sample.

**Figure 4 materials-12-01423-f004:**
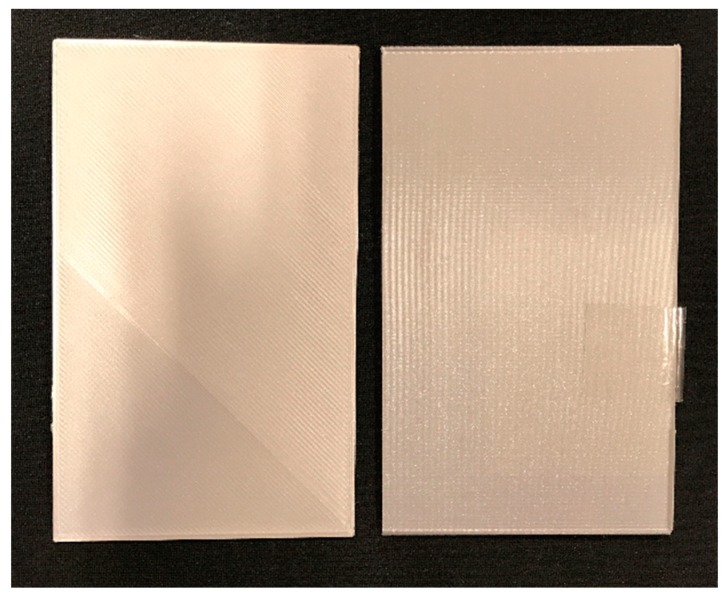
Left: the 3D object surface when printed in the XY plane. Right: the 3D object surface when printed in the Z plane.

**Figure 5 materials-12-01423-f005:**
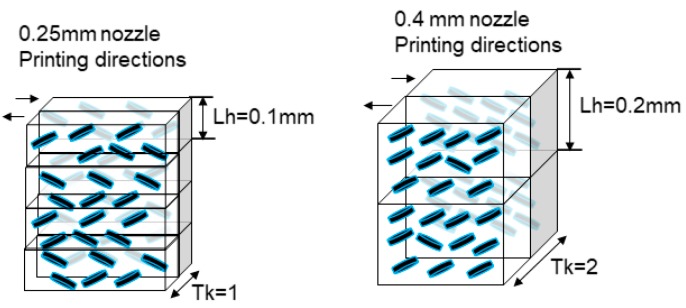
Schematic of the hypothetical orientation of the flakes: left in the sample thickness 0.5 mm (Tk = 1) and layer height (Lh) 0.1 mm, right sample thickness 0.8 mm (Tk = 2) and layer height (Lh) 0.2 mm.

**Figure 6 materials-12-01423-f006:**
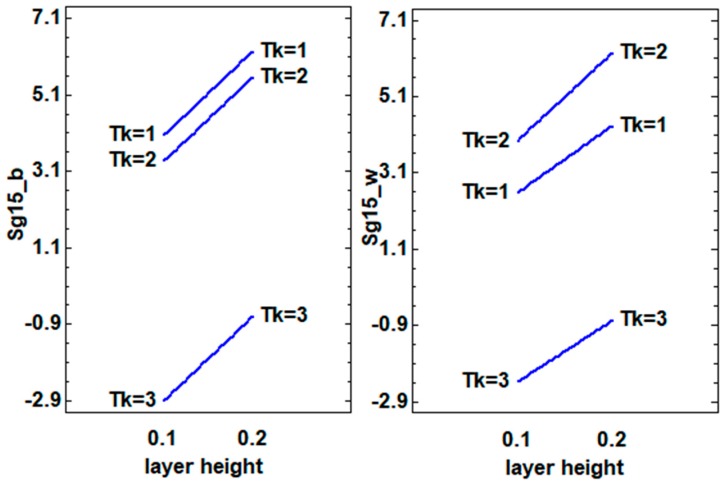
Interaction plots (layer height (B) and sample thickness (C) factors) for the sparkle values at 15° on white (_w) and black (_b) backgrounds.

**Figure 7 materials-12-01423-f007:**
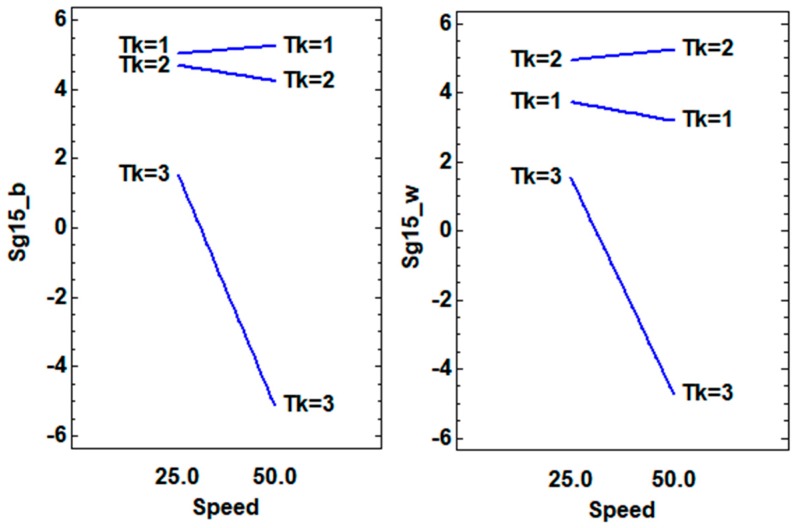
Interaction plots (printing speed (A) and sample thickness (C) factors) for the sparkle values at 15° on the white (_w) and black (_b) backgrounds.

**Figure 8 materials-12-01423-f008:**
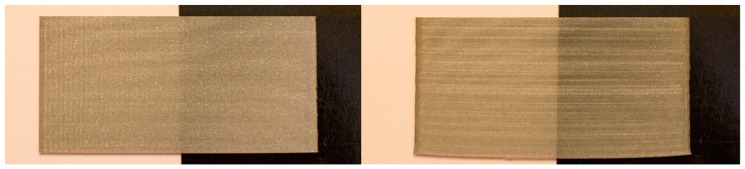
Narrow thickness samples printed with PLA filaments (left) and ABS filaments (right).

**Figure 9 materials-12-01423-f009:**
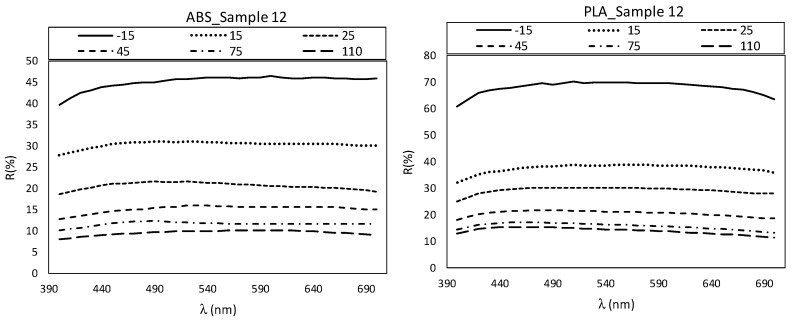
Visible reflectance spectrum values from PLA and ABS sample 12 in DoE conditions.

**Table 1 materials-12-01423-t001:** Results of the PLA (poly(lactic acid)) samples when changing printing parameters: printing speed (Sp), layer height (Lay), sample thickness (Tk). Sparkle (Sg), graininess (G) and flop were measured. The sparkle measurements were taken at different geometries (15°, 45° and 75°) on the white (w) and black (b) backgrounds.

Sp	Lay	Tk	Sg45_b ^1^	Sg45_w	Sg15_b	Sg15_w	Sg75_b	Sg75_w	G_b	G_w	Flop_b	Flop_w
25	0.2	0.5	1.34	1.38	2.89	2.77	1.14	1.17	5.93	5.67	3.28	3.02
25	0.2	0.8	1.46	1.52	2.98	2.65	0.71	0.70	9.27	9.27	3.15	3.14
25	0.1	1.6	1.59	1.51	2.69	2.94	0.74	0.80	7.32	7.30	3.31	3.31
50	0.1	0.8	1.89	1.60	2.72	2.73	1.03	0.84	6.50	6.58	2.56	2.54
25	0.2	0.5	1.38	1.20	3.34	2.95	0.80	0.70	5.48	4.93	3.06	2.76
50	0.2	1.6	1.24	1.34	2.56	2.78	0.51	0.56	8.97	8.96	3.63	3.63
50	0.2	0.5	1.37	1.51	3.02	2.82	1.35	1.04	6.80	6.47	3.19	2.96
50	0.1	0.5	1.67	1.35	3.43	3.07	0.82	0.64	5.91	5.32	3.19	2.84
50	0.2	0.8	1.46	1.06	2.74	2.52	0.70	0.47	9.30	9.34	3.05	3.05
25	0.2	1.6	1.44	1.46	2.96	2.93	0.85	0.78	9.31	9.32	3.29	3.31
50	0.1	1.6	1.80	1.75	3.25	3.29	0.71	0.60	7.37	7.39	3.51	3.51
25	0.1	0.8	2.11	1.75	3.03	3.14	0.97	1.04	7.07	7.04	3.01	2.96

^1^ All the measurements taken on black (b) and white (w) backgrounds.

**Table 2 materials-12-01423-t002:** Results of the ABS samples when changing printing parameters: printing speed (Sp), layer height (Lay), sample thickness (Tk). Sparkle (Sg), graininess (G) and flop were measured. The sparkle measurements were taken at different geometries (15°, 45° and 75°) on the white (w) and black (b) backgrounds.

Sp	Lay	Tk	Sg45_b ^1^	Sg45_w	Sg15_b	Sg15_w	Sg75_b	Sg75_w	G_b	G_w	Flop_b	Flop_w
15	0.2	2,4	4.39	4.52	5.15	4.97	2.63	2.57	4.63	4.66	3.52	3.51
15	0.2	0,8	6.05	4.54	6.22	4.75	3.49	2.80	5.14	3.74	3.55	2.97
15	0.1	1,6	4.48	4.07	4.07	3.96	3.73	3.66	5.20	4.97	5.14	5.04
10	0.1	0,8	3.99	3.08	3.81	3.09	3.98	3.23	4.70	3.72	4.94	3.83
15	0.1	2,4	3.22	3.23	3.27	3.16	3.35	3.39	4.20	4.09	4.43	4.35
10	0.2	1,6	5.76	5.42	6.55	6.21	4.55	4.68	5.13	5.00	3.54	3.45
10	0.2	2,4	5.06	5.16	6.46	6.03	3.10	3.26	5.05	5.03	3.25	3.26
10	0.1	2,4	2.93	2.92	4.63	4.61	1.77	1.71	4.75	4.72	4.92	4.87
10	0.2	0,8	6.32	4.36	5.98	5.02	3.43	2.48	5.38	4.18	3.70	2.95
15	0.2	1,6	5.58	5.76	5.67	5.68	2.78	3.03	4.81	4.92	3.56	3.57
10	0.1	1,6	3.66	3.59	3.37	3.31	3.70	3.65	4.36	4.27	4.43	4.35
15	0.1	0,8	4.71	3.94	3.66	3.14	3.97	3.40	5.11	4.12	5.31	4.65

^1^ All the measurements taken on black (b) and white (w) backgrounds.

**Table 3 materials-12-01423-t003:** Variance analysis summary for the PLA samples with Sg45_w as the optimization response and the factors printing speed (A), layer height (B) and sample thickness (C).

Factor	Sum of Squares	^(a)^ df	Medium Square	F-Ratio	P-Value
B	0.066008	1	0.066008	9.64799	0.05
C	0.053450	2	0.026725	3.90621	0.15
AB	0.039675	1	0.039675	5.79903	0.10
AC	0.112550	2	0.056275	8.22533	0.06
BC	0.164017	2	0.082008	11.9866	0.04

R-squared = 95.5011%. ^(a)^ Degrees of freedom: the number of independent values or quantities which can be assigned to a statistical distribution.

**Table 4 materials-12-01423-t004:** Variance analysis summary for the PLA samples with Sg75_w as the optimization response and the factors printing speed (A), layer height (B) and sample thickness (C).

Factor	Sum of Squares	df	Medium Square	F-Ratio	P-Value
A	0.090133	1	0.090133	515.048	0.0019
B	0.000833	1	0.000833	4.7619	0.1608
C	0.083517	2	0.041758	238.619	0.0042
AB	0.001200	1	0.001200	6.85714	0.1201
AC	0.009217	2	0.004608	26.3333	0.0366
BC	0.315317	2	0.157658	900.905	0.0011

R-squared = 99.9301%.

**Table 5 materials-12-01423-t005:** The design of experiment (DoE) analysis summary of the PLA samples.

Response	Significant Factor and Best Level
Sg45_b	B (0.1 mm)
Sg45_w	BC (0.1–0.8 mm)
Sg15_b	-
Sg15_w	B (0.1 mm)
Sg75_b	BC (0.1–0.5 mm)
Sg75_w	A (25 mm/s^−1^), BC (0.2–0.5 mm)
G_b	BC (0.2–1.6 mm)
G_w	BC (0.2–1.6 mm)
Flop_b	C (1.6 mm)
Flop_w	C (1.6 mm)

Factors: Printing speed (A), Layer height (B), Sample thickness (C).

**Table 6 materials-12-01423-t006:** Variance analysis summary of the PLA samples with Sg45_b as the optimization response and the factors printing speed (A), layer height (B) and sample thickness (C).

Factor	Sum of Squares	df	Medium Square	F-Ratio	P-Value
A	0.042008	1	0.042008	387.77	0.0026
B	0.308291	1	0.308291	2845.8	0.0004
C	0.320095	2	0.160048	1477.4	0.0007
AB	0.725208	1	0.725208	6694.2	0.0001
AC	0.147117	2	0.073558	679	0.0015
BC	0.030650	2	0.015325	141.46	0.007

R-squared = 99,9984%.

**Table 7 materials-12-01423-t007:** The DoE analysis summary of the ABS samples.

Response	Significant Factor and Best Level
Sg45_b	A.B.C (50mm/s^−1^, 0.1 mm, 0.8–0.6 mm)
Sg45_w	B (0.2 mm)
Sg15_b	B (0.2 mm)
Sg15_w	B (0.2 mm)
Sg75_b	B (0.1 mm)
Sg75_w	-
G_b	AB (50 mm/s^−1^. 0.1 mm)
G_w	-
Flop_b	B (0.1 mm)
Flop_w	B (0.1 mm)

Factors: Printing speed (A), Layer height (B), Sample thickness (C).

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
