# Peer review of "Maximization of FDM-3D-Objects Gonio-Appearance Effects Using PLA and ABS Filaments and Combining Several Printing Parameters: “A Case Study”"

_materials, 2019, doi:10.3390/ma12091423_

Round 1
Reviewer 1 Report
The material is commercial and the analyzing method is not convincible. It would be better to put some scientific analyze. There are some other questions which should be addressed:
what kind of 3D printing pattern affected the results and what is the metal morphology under different printing temperature?
2. How many percentages of the metal in the plastic could provide the best printing and good enough mechanical strength?
3. How about sintering the products and what will happen to the products?
Author Response
Comments and Suggestions for Authors
The material is commercial and the analyzing method is not convincible. It would be better to put some scientific analyze. There are some other questions which should be addressed:
In this work the aim was to work with commercial filaments in order to get the maximum textural effects of the 3D objects under real working conditions. On the one hand the scientific approach and analysis were statistical analysis using multivariate methods to design efficiently the number of experiments, and find the combined effects of different printing parameters in the final visual appearance. On the other hand, the visual appearance characterization was made using the colour science. In this way, a multiangle spectrophotometer was used to perform the measurements both on color and texture (sparkle and graininess). These kind of materials (gonio-apparent materials) are not new on the automotive or cosmetic industry, however, they are still under review due to their complexity. For instance, there are different technical committee of the International Commission on Illumination (CIE) that are working in this topic (JTC-12, 8-17). To reinforce, the scientific quality of the work, we added more statistical details and equations for the textural effects calculation. Furthermore, we improved the results discussion by considered the reviewer suggestions and questions.
what kind of 3D printing pattern affected the results and what is the metal morphology under different printing temperature?
As we specify and clarify in the revised paper, there is no 3D pattern in the objects that we printed, because we did not use the infill option. In all the samples, the infill parameter was 0% and we increase the shell thickness in order to get a solid sample without gaps. The metal morphology was only influenced by the filament thickness and the deposition direction. We added a figure with the printing factors scheme to clarify the controlled and changed parameters and the metallic/diffractive pigments possible disposition across the printed sample. In addition, we decided to use this scheme as graphical abstract of this paper. This is a good question to improve our paper, therefore, thank you very much for that question.
In the new version L148: The infill parameter was 0%, but samples were printed with no gap using shell thickness to fill samples. With this method there was no pattern in the infill of any sample, in all samples there were only a change in the direction of the filament deposition during the sample printing. The metallic or diffractive pigments orientation depends on the pigment size and shape and the layer height (Figure 3). A multifactorial DOE 22·31 was selected for each material.
2. How many percentages of the metal in the plastic could provide the best printing and good enough mechanical strength?
As the reviewer said, in this work, we used commercial filaments, and the metal concentration in both filaments is a protected information from the filament suppliers. As we had no control over the metal concentration in both filaments, we are not able to answer about the influence of the metal content in this work. However, with this method, we find the influence about the printing parameters that we can change and control and we provide the method to increase the textural desirable effects with filaments that contains metallic and diffractive pigments.
3. How about sintering the products and what will happen to the products?
Our hypothesis is that with the sintering, the textural effects are going to be complete different. The final surface is completely different because the printing method is completely different, and the effect pigments orientation and deposition are going to be different to. Depending on the sintering temperatures, the effect pigments structures could be also affected. We suppose that the diffractive effect is going to be also more difficult to control, because of the different pigment orientations in the sintered particles. The same could be expected with other 3D printing technologies as DLP for instance. In all different 3D printing technologies, it should be investigated the printing parameters effects in the final object appearance.

Reviewer 2 Report
This manuscript conducts a design of experiment to study the effects of printing parameters on sparkle, graininess, and flop of two materials, PLA and ABS. This study is straight forward but very specific to the given materials and printer. It reads more like a technical report than a paper. The authors should include more scientific insight or discussion on results in addition to the data.
Specific comments are listed below:
1. The DoE include three independent variables, printing speed, layer height, and sample thickness. It is unclear why thickness is a variable? Intuitively, the sample size should be controlled in order to see how the printing parameters (speed and layer height) affect the sparkle and graininess. As can be seen, the results are sensitive to sample size (thickness), so it should be a control variable.
2. It needs to be clarified how three different angles (15˚, 45˚, and 75 ˚) are defined with respect to the sample. And what do they actually mean? A drawing or actual picture may be helpful.
3. All the measurements (Sg, G, and Flop) have no unit. please confirm.
4. In Figure 3, the effect on layer height turns into the opposite trend when TK = 1. There needs to be an explanation for such a strong interaction — same thing for Figure 4. Also, Figs 3 and 4 are not particularly discussed in the text.
5. This manuscript does not have a Discussion section, which is uncommon for a technical paper.
6. “Gonio-appearance” does not seem to be a common technical term to my understanding. Please confirm or provide a reference for it.
Author Response
This manuscript conducts a design of experiment to study the effects of printing parameters on sparkle, graininess, and flop of two materials, PLA and ABS. This study is straight forward but very specific to the given materials and printer. It reads more like a technical report than a paper. The authors should include more scientific insight or discussion on results in addition to the data.
The reviewer is right, for this reason, we improved scientific discussion (blue type in new paper version). It is important to emphasize that the aim of this paper is to control the final textural and gonio-apparent effects in the 3D objects since there are no standards to control the effects in 3D objects from these technologies and the development of new filament has grown the interest in last years. However, there are no guarantees that the filaments with effects or diffractive pigments, keep this effects in the final printed objects. This work establishes an efficient and scientific methodology to control the 3D objects appearance when it is only possible to control the printing parameters.
Specific comments are listed below:
1. The DoE include three independent variables, printing speed, layer height, and sample thickness. It is unclear why thickness is a variable? Intuitively, the sample size should be controlled in order to see how the printing parameters (speed and layer height) affect the sparkle and graininess. As can be seen, the results are sensitive to sample size (thickness), so it should be a control variable.
The reviewer is right again. We control two parameters directly related to the 3D printer: speed and layer height. However, another important factor which can affect the final visual appearance is the sample size as the reviewer said. This factor/variable is not directly related to any parameter from the 3D printer, however, from the results, it is evident that it influence on the visual appearance, for this reason, this variable was considered as an independent variable in order to find the simple effect of the sample size increment in the final 3D objects appearance, and control also the interaction between this factor and the printing speed or layer height. We added these comments in the methodology (and new Figure 3 commented an added before) to let clear the different kind of parameters that we selected and control in this work. Furthermore, sample thickness is a critical variable to get translucent or opaque samples. Depending on the filament material properties and the sample thickness, the measurements over white and black background were the same or not, and consequently we got different responses increments depending of the sample thickness.
In addition, we add in the new version the blue comments: A statistical design of the experiment (DoE) was used to select the parameter values that had to be selected for each sample. F-ratio was used to calculate the P-Value to know the significance level of the difference in the measured responses under different experiment conditions. The entire statistical test was made at 95% of confidence level. For PLA, the printing parameters of printing temperature were set at 220ºC and bed temperature at 70ºC. Then the following factors and levels were selected for the experiment: printing speed [25-50] mm·s-1, layer height [0.1-0.2] mm, sample thickness [0.5, 0.8 and 1.6] mm.
2. It needs to be clarified how three different angles (15˚, 45˚, and 75 ˚) are defined with respect to the sample. And what do they actually mean? A drawing or actual picture may be helpful.
The BYK‐mac multiangle spectrophotometer is capable of measuring spectral reflectance at six geometries for color characterization, three geometries for sparkle, and one geometry for graininess characterization simultaneously. This device characterizes color by measuring the sample at different aspecular angles: ‐15, 15, 25, 45, 75, and 110°. Following CIE standards, these geometries are represented as 45°x:‐60°, 45°x:‐30°, 45°x:‐20°, 45°x:0°, 45°x:30°, and 45°x:65° respectively. For sparkle. this device also performs the measurements in three different measurement geometries. In this way, the illumination angle is measured from the perpendicular direction to the sample (from 15º, 45º and 75º, as can be seen in the next figure). The observation angle is always in the perpendicular direction to the sample (see next figure). Thus, following CIE standards, these geometries are represented as 15°x:0°, 45°x:0°, and 75°x:0° respectively (Fig. 2). Sparkle effect is dependent on the measurement geometry due to the different particle/flake orientation. Thus, by measuring at different measurement geometry, the contribution of particles with different orientation is evaluated. However, graininess, G, is a texture effect defined for diffuse illumination conditions. For this reason, it is not mentioned any specific measurement geometry. To clarify these measurement geometries for readers, we added a new paragraph and a new figure (Figure 2). Also we add new reference for this topic:
[24] E. Chorro et al., “The minimum number of measurements for colour, sparkle, and graininess characterisation in gonio-apparent panels,” Color. Technol., vol. 131, no. 4, pp. 303–309, Aug. 2015.
3. All the measurements (Sg, G, and Flop) have no unit. please confirm.
Yes, this is correct. Sg and G are parameters obtained after digital image processing by considering the digital grey levels and unknown specific calculations (it is a protected information from BYK Gardner), therefore no units are defined. The flop is a parameter that measures the difference in lightness between measurement geometries according to the L* variable of the CIELAB color space. Lightness evaluates the luminance (Y, cd/m2) of a sample judged in relation to the luminance (Yw, cd/m2) of a reference white. Then, L* has no unit, and consequently, Flop has no unit.
4. In Figure 3, the effect on layer height turns into the opposite trend when TK = 1. There needs to be an explanation for such a strong interaction — same thing for Figure 4. Also, Figs 3 and 4 are not particularly discussed in the text.
We added the following discussion to explain that interaction:
Other works find that the spatial orientation has large impact on Build Time in FDM process [25], and temperature and time could affect to the orientation of the metallic particles across the printed surface. In a specific research, it was checked that the final surface quality in FDM printed objects is very sensitive to different printing factors as layer height and the sample surface angle. For instance, the less roughness was achieved using the lower layer thickness and the maximum printing angle [26]. Moreover, in metallic coatings, flakes tend to orient themselves parallel to the substrate. The physical mechanism behind this orientation effect is mainly related to the film shrinkage during solvent evaporation, and that effect will be similar due to layer shrinkage by the temperature decrease and the several layer depositions in the printed object. However, flake orientation also depends on the flake diameter [27] and flake concentration or pigment type. Silver dollar metallic flakes orient much better than cornflake metallic, and coarse metallic flakes orient much better than fine metallic flakes, and large flakes orient themselves much better than small flakes [28]. However, the colour formulation (composition) of the filament is not available in commercial samples.
Due to the printing directions, the orientation of the pigment flakes could be as it is shown in Figure 5. When it is increased the layer height the probability to find pigment flakes with a good orientation increase, improving the Sg_15 effect. The Sg_15 data are stronger than Sg_45 and Sg_75 and represents the sparkle effect due to the good orientation of the pigment flakes, in this way, it is achieving the objective of this work, to maximize this effect. The Sg_15 effect is higher in black background because of the contrast increment, as expected. At lowest thickness (Tk=1 or Tk=2) the orientation of the flakes is improved, and it also increases with the layer height, because it is possible to get more concentration of well orientated particles (Figure 6). This behaviours and trends were more difficult to find in Sg_45 or Sg_75 results because of in this measurement conditions the sparkle measured mainly come from the disorientated flakes. For that reason, we replaced Figure 3 and 4 with New Figures 6 and 7.
As we discussed before, speed can affect to the preferred orientation of the pigment flakes. The flakes show better orientation with thinner samples and lower speed over white or black background. The lower printing speed allows the orientation of the pigment flakes, and this phenomenon is well detected with the samples with more thickness (Tk=3). At lower thickness (Tk=1 or Tk=2) the good orientation was achieved with the decrease of the material content and it is possible to increase the printing speed without significant decrease of the sparkle values (Figure 7).
5. This manuscript does not have a Discussion section, which is uncommon for a technical paper.
We added the discussion (including the new discussions suggested by the reviewers) throughout the results section. All changes are highlighted in blue in the new paper version.
6. “Gonio-appearance” does not seem to be a common technical term to my understanding. Please confirm or provide a reference for it.
This term is common inside the topic about special effect pigments. For instance, in the added reference title you can find the term: “The minimum number of measurements for colour, sparkle, and graininess characterisation in gonio-apparent panels”.
In addition, it is possible to find more scientific papers with this terminology:
- A new goniospectrophotometer for measuring gonio‐apparent materials, Coloration Technology, 121(2), 96-103, 2005
- Model‐based corrections of geometric errors in multiangle measurements of gonio‐apparent coatings, Color Research and Application, 41(4), 372-383, 2016.
- Automatic gonio-spectrophotometer for the absolute measurement of the spectral BRDF at in- and out-of-plane and retroreflection geometries, Metrologia, 49(3), 2012.

Reviewer 3 Report
The paper provides a systematic approach to find out the best printing parameters of FDM for two different commercial filaments (PLA and ABS). I think that there is no problem to publish as it is.
Author Response
Thank you very much. He hope that you enjoy the new version as well.
Round 2
Reviewer 1 Report
The author should provide more scientific analyze like simulation or provide some novelty of 3D printing or put the material in some applications. Please check the following literature:
3D-printed flexible polymer stents for potential applications in inoperable esophageal malignancies.
Author Response
REV_1
Comments and Suggestions for Authors
The author should provide more scientific analyze like simulation or provide some novelty of 3D printing or put the material in some applications. Please check the following literature:
3D-printed flexible polymer stents for potential applications in inoperable esophageal malignancies.
The reviewer is right; other researchers used simulation software to select the optimal parameters to develop new 3D printed objects. Other example are “Sensitivity of RP surface finish to process parameter variation” that we already cited and commented in the new introduction version. They used AutoCAD© for the sample design, and we used free software blender for our sample designs, as we also add in the method description. In addition, we can find also case studies as the one that we presented “Effect of varying spatial orientations on build time requirements for FDM process: A case study.” We change the actual title to let clear that this is a case study as well: Maximisation of FDM-3D-objects gonio-appearance effects using PLA and ABS filaments and combining several printing parameters: “a case study”.
In this work we cannot control the filament parameters, because the interested companies in the development of new filaments with textural effects, do not provide the filament descriptions as the effect pigment properties (shape, composition), or effect pigment concentration. The company provide us the filament with diffractive pigments with no information, with the aim to characterize the final metallic, sparkle and flop effects that we could achieve with a FDM printer. We emphasize that in the paper and add an example with the reflectance values of the ABS and PLA samples (Figure 9) to demonstrate one of our main findings which is that the diffractive effect was lost after the printing process. This finding is remarkable and must be taken into account by the filament designers, before selling filaments with supposed properties, which they are not going to achieve then. In addition, we provide a scientific and statistical method to get the best final properties in 3D-pinted objects with textural effects, like metallic effect (Flop values), graininess and sparkle. The automotive industry and the 3D filament industries are going to be the most interested in the facts that we proved in this case study. All our improved discussions and findings are highlighted in blue in the text. You can find the improved Introduction, discussion, results and conclusions as you asked. We hope that you find interesting and ready for publication our work.
In the new paper version:
“Furthermore, any filament producer can guarantee final textural effects on the produced 3D printer object. The critical factors in FDM printers have been studied to improve the mechanical properties of the 3D object or reduce printing costs [16]–[20]. Vasudevarao et al. [21] proposed an experimental design to determine significant factors and their interactions for optimal surface finish of parts fabricated via Fused Deposition Modelling process. As Rapid Prototyping is moving towards Rapid Manufacturing there is an increasing stress on obtaining good quality parts. Quality of a prototype includes the surface quality, the mechanical strength and dimensional accuracy among other things. Surface Finish is critical not only for better functionality and look, but also for cost reduction in terms of reduced post-processing of parts (this includes sanding, filing etc.) and overall prototyping time reduction also. However, the effect of 3D printing parameters on the final visual appearance has not yet been analysed. The aim of this work was to control the metallic and interference effect on 3D-printed objects by changing certain printing parameters, such as filament material, sample thickness, printing speed, etc.
2. Materials and Methods
We used the free software platform Blender for the 3D objects designs. To build the 3D samples, an Ultimaker 2+ 3D was used. This printer machine is capable of working with different nozzles sizes: 0.25, 0.4, 0.6 and 0.8 mm. This machine works with a 2.85 mm filament diameter. However, changing printing parameters is possible to work with different filament diameters. In addition we used CURA software, designed by Ultimaker (Umetani, Bickel, Matusik, 2015), to change the sample and 3D printing parameters.
For this study, work was done with two commercial filaments from different polymers and with distinct properties. PLA grey and metallic filaments were used (2.85 mm in diameter), obtained from Utimaker (UM9015 PLA Silver Metallic RAL 9006), along with ABS grey with 3% diffractive pigments (MAXITHEN® SAN702767/10 Metallic Silver), and a diameter of 1.75 mm, which came from Schlenk Metallic Pigments GmbH (Figure 1). The last filament was not available in the market before our test. Schlenk Company was interested to know the diffractive pigments behaviour under work conditions. This work is going to provide critical information to the filament makers, as the final effects limits to this new filaments.
Results and discussion_ line 296:
The most significant factor to achieve high flop values and Sg75_b was layer height (0.1mm). The lower the layer height, the higher the flop value on the black or white background (Table 7). This may be due to the diffractive pigment orientations in thinner layers. However, this parameter did not suffice to maintain the diffractive effect, and only in sparkle was it possible to see different colours change the direct or diffuse illumination. The textural and final metallic effects were the same in both samples PLA and ABS, as seen in Figure 8. The visual appearance must change significantly with the orientation angle of measurement, mostly in the ABS filament with diffractive pigments. However, the “rainbow” effect disappears in working conditions. The reflectance spectrum values in VIS range change mainly in parallel as the PLA samples with metallic pigments, and is not possible to observe significant differences in the band positions or the spectrum shapes with the change of the orientation measurement [23]. That means mainly changes in L*ab value due to the orientation measurement changes, but no difference colours can be observed due to the diffractive pigments, as its represented in the example (Figure 9). The effect that was firstly observed in the filament was lost in the melting process of 3D printing. The different orientations of layers during the printing process did not allows the diffraction effect to be controlled and maintained as in the raw material.
Figure 9. VIS reflectance spectrum values from PLA and ABS sample in 12 DoE conditions
4. Conclusions
The aim of this work was to find significant differences in the textural effect of 3D-printed objects using different filament materials with metallic and diffractive pigments, and to discover the optimal conditions that achieve the maximum sparkle, flop and graininess effects. In addition, a filament producer company Schlenk, was interested to know the diffractive pigments behaviour under work conditions for the filament prototype that they provide for this work. As pigments manufacturers they are focused in innovation to develop coatings, and materials interesting for advanced applications as automotive sector. With the multivariate factorial DoE, it was possible to define the most significant printing factors for both filaments ABS prototype and PLA with metallic pigments, improving the gonio-appearance of the 3D-printed objects. The best conditions to increase the metallic effects of the PLA objects at the 15º observation angle are achieved at a low layer height (0.1 mm) and low sample thickness. In addition, it is recommended to use low printing speed to allow the effect pigments particles orientation. The graininess effect is stronger when layer height is increased (0.2 mm), so working at high layer levels is recommended if increasing the diffuse coarseness surfaces is important. However, this measurement could be affected for the surface roughness of FDM samples, and this should be taken into account. For this reason, sparkle and flop parameters are better to analyse the printing factor effects with FDM 3D objects. To increase the flop parameter (on a black or a white background), increasing sample thickness is recommended.
In addition, to provide our scientific and systematic work, we attached a supplementary material with all the statistical analysis that we have make after the discussion and conclusions of this work.
SUPPLEMENTARY MATERIAL: ANOVA TEST AND INTERACTION PLOTS
ABS
Sg45 BLK
Effects analysis
Cat. Factors:
C=Thickness
Quant. Factors:
A=Speed
B=layer height
Source | Sum of Squares | f.d. | Medium Square | F-Ratio | P-Value |
A | 0,0420083 | 1 | 0,0420083 | 387,769 | 0,0026 |
B | 0,308291 | 1 | 0,308291 | 2845,76 | 0,0004 |
C | 0,320095 | 2 | 0,160048 | 1477,36 | 0,0007 |
AB | 0,725208 | 1 | 0,725208 | 6694,23 | 0,0001 |
AC | 0,147117 | 2 | 0,0735583 | 679,0 | 0,0015 |
BC | 0,03065 | 2 | 0,015325 | 141,462 | 0,0070 |
R = 99,9984%
Sg45 WHT
Effects analysis
Cat. Factors:
C=Thickness
Quant. Factors:
A=Speed
B=layer height
Source | Sum of Squares | f.d. | Medium Square | F-Ratio | P-Value |
A | 0,195075 | 1 | 0,195075 | 1,58141 | 0,2641 |
B | 6,64541 | 1 | 6,64541 | 53,8722 | 0,0007 |
C | 1,46622 | 2 | 0,733108 | 5,94308 | 0,0477 |
BC | 0,451017 | 2 | 0,225508 | 1,82812 | 0,2536 |
R = 93,4207%
Sg15 BLK
Effects analysis
Cat. Factors:
C=Thickness
Quant. Factors:
A=Speed
B=layer height
Source | Sum of Squares | f.d. | Medium Square | F-Ratio | P-Value |
A | 0,6348 | 1 | 0,6348 | 3,75592 | 0,1104 |
B | 14,564 | 1 | 14,564 | 86,1709 | 0,0002 |
C | 1,15962 | 2 | 0,579812 | 3,43057 | 0,1154 |
AC | 1,15755 | 2 | 0,578775 | 3,42443 | 0,1157 |
R= 95,0884%
Sg15 WHT
Effects analysis
Cat. Factors:
C=Thickness
Quant. Factors:
A=Speed
B=layer height
Source | Sum of Squares | f.d. | Medium Square | F-Ratio | P-Value |
A | 0,567675 | 1 | 0,567675 | 4,13632 | 0,1349 |
B | 10,811 | 1 | 10,811 | 78,7735 | 0,0030 |
C | 0,957596 | 2 | 0,478798 | 3,48872 | 0,1649 |
AC | 1,02305 | 2 | 0,511525 | 3,72718 | 0,1537 |
BC | 0,266217 | 2 | 0,133108 | 0,969883 | 0,4733 |
R = 97,173%
Sg75 BLK
Effects analysis
Cat. Factors:
C=Thickness
Quant. Factors:
A=Speed
B=layer height
Source | Sum of Squares | f.d. | Medium Square | F-Ratio | P-Value |
B | 1,21157 | 1 | 1,21157 | 5,83945 | 0,0604 |
C | 0,802248 | 2 | 0,401124 | 1,93331 | 0,2388 |
AB | 1,1907 | 1 | 1,1907 | 5,73887 | 0,0620 |
AC | 1,03752 | 2 | 0,518758 | 2,50028 | 0,1768 |
R = 82,4463%
Sg75 WHT
Effects analysis
Cat. Factors:
C=Thickness
Quant. Factors:
A=Speed
B=layer height
Source | Sum of Squares | f.d. | Medium Square | F-Ratio | P-Value |
A | 0,00213333 | 1 | 0,00213333 | 0,00505399 | 0,9498 |
B | 1,25624 | 1 | 1,25624 | 2,9761 | 0,2266 |
C | 0,693334 | 2 | 0,346667 | 0,821275 | 0,5491 |
AB | 1,25453 | 1 | 1,25453 | 2,97206 | 0,2269 |
AC | 0,975317 | 2 | 0,487658 | 1,15529 | 0,4640 |
BC | 0,624817 | 2 | 0,312408 | 0,740114 | 0,5747 |
R = 85,8947%
G BLK
Effects analysis
Cat. Factors:
C=Thickness
Quant. Factors:
A=Speed
B=layer height
Source | Sum of Squares | f.d. | Medium Square | F-Ratio | P-Value |
A | 0,00653333 | 1 | 0,00653333 | 0,0619127 | 0,8267 |
B | 0,185149 | 1 | 0,185149 | 1,75455 | 0,3164 |
C | 0,352667 | 2 | 0,176333 | 1,67101 | 0,3744 |
AB | 0,2352 | 1 | 0,2352 | 2,22886 | 0,2740 |
AC | 0,303517 | 2 | 0,151758 | 1,43813 | 0,4102 |
BC | 0,0193167 | 2 | 0,00965833 | 0,0915265 | 0,9161 |
R = 85,0633%
G WHT
Effects analysis
Cat. Factors:
C=Thickness
Quant. Factors:
A=Speed
B=layer height
Source | Sum of Squares | f.d. | Medium Square | F-Ratio | P-Value |
A | 0,0147 | 1 | 0,0147 | 0,153712 | 0,7328 |
B | 0,118021 | 1 | 0,118021 | 1,2341 | 0,3823 |
C | 0,355874 | 2 | 0,177937 | 1,86062 | 0,3496 |
AB | 0,154133 | 1 | 0,154133 | 1,61171 | 0,3320 |
AC | 0,3318 | 2 | 0,1659 | 1,73475 | 0,3657 |
BC | 0,0866667 | 2 | 0,0433333 | 0,45312 | 0,6882 |
R = 92,7219%
Flop BLK
Effects analysis
Cat. Factors:
C=Thickness
Quant. Factors:
A=Speed
B=layer height
Source | Sum of Squares | f.d. | Medium Square | F-Ratio | P-Value |
A | 0,0444929 | 1 | 0,0444929 | 0,404272 | 0,5701 |
B | 5,40393 | 1 | 5,40393 | 49,1013 | 0,0060 |
C | 0,123023 | 2 | 0,0615116 | 0,558908 | 0,6218 |
AC | 0,110493 | 2 | 0,0552464 | 0,501981 | 0,6486 |
BC | 0,0396714 | 2 | 0,0198357 | 0,180231 | 0,8435 |
R = 94,6489%
Flop WHT
Effects analysis
Cat. Factors:
C=Thickness
Quant. Factors:
A=Speed
B=layer height
Source | Sum of Squares | f.d. | Medium Square | F-Ratio | P-Value |
A | 0,155358 | 1 | 0,155358 | 1,96926 | 0,2195 |
B | 4,56695 | 1 | 4,56695 | 57,8888 | 0,0006 |
C | 0,180596 | 2 | 0,0902981 | 1,14458 | 0,3897 |
AC | 0,199523 | 2 | 0,0997616 | 1,26454 | 0,3594 |
R= 93,295
PLA
Sg45 BLK
Effects analysis
Cat. Factors:
C=Thickness
Quant. Factors:
A=Speed
B=layer height
Source | Sum of Squares | f.d. | Medium Square | F-Ratio | P-Value |
A | 0,00100833 | 1 | 0,00100833 | 0,0425906 | 0,8497 |
B | 0,378075 | 1 | 0,378075 | 15,9694 | 0,0281 |
C | 0,18035 | 2 | 0,090175 | 3,80887 | 0,1502 |
AC | 0,0367167 | 2 | 0,0183583 | 0,775431 | 0,5352 |
BC | 0,06845 | 2 | 0,034225 | 1,44562 | 0,3634 |
R = 90,3449%
Sg45 WHT
Effects analysis
Cat. Factors:
C=Thickness
Quant. Factors:
A=Speed
B=layer height
Source | Sum of Squares | f.d. | Medium Square | F-Ratio | P-Value |
B | 0,0660083 | 1 | 0,0660083 | 9,64799 | 0,0530 |
C | 0,05345 | 2 | 0,026725 | 3,90621 | 0,1461 |
AB | 0,039675 | 1 | 0,039675 | 5,79903 | 0,0952 |
AC | 0,11255 | 2 | 0,056275 | 8,22533 | 0,0606 |
BC | 0,164017 | 2 | 0,0820083 | 11,9866 | 0,0371 |
R = 95,5011%
Sg15 BLK
Effects analysis
Cat. Factors:
C=Thickness
Quant. Factors:
A=Speed
B=layer height
Source | Sum of Squares | f.d. | Medium Square | F-Ratio | P-Value |
A | 0,00240833 | 1 | 0,00240833 | 0,0280338 | 0,8824 |
B | 0,143008 | 1 | 0,143008 | 1,66466 | 0,3260 |
C | 0,24605 | 2 | 0,123025 | 1,43205 | 0,4112 |
AB | 0,0602083 | 1 | 0,0602083 | 0,700844 | 0,4906 |
AC | 0,0917167 | 2 | 0,0458583 | 0,533805 | 0,6520 |
BC | 0,0862167 | 2 | 0,0431083 | 0,501795 | 0,6659 |
R = 78,5611
Sg15 WHT
Effects analysis
Cat. Factors:
C=Thickness
Quant. Factors:
A=Speed
B=layer height
Source | Sum of Squares | f.d. | Medium Square | F-Ratio | P-Value |
B | 0,226875 | 1 | 0,226875 | 10,5851 | 0,0313 |
C | 0,10365 | 2 | 0,051825 | 2,41796 | 0,2049 |
AC | 0,0877167 | 2 | 0,0438583 | 2,04627 | 0,2443 |
BC | 0,00945 | 2 | 0,004725 | 0,220451 | 0,8113 |
R = 83,3017%
Sg75 BLK
Effects analysis
Cat. Factors:
C=Thickness
Quant. Factors:
A=Speed
B=layer height
Source | Sum of Squares | f.d. | Medium Square | F-Ratio | P-Value |
B | 0,00300833 | 1 | 0,00300833 | 0,282546 | 0,6319 |
C | 0,211667 | 2 | 0,105833 | 9,93999 | 0,0475 |
AB | 0,00300833 | 1 | 0,00300833 | 0,282546 | 0,6319 |
AC | 0,0474 | 2 | 0,0237 | 2,22593 | 0,2554 |
BC | 0,275267 | 2 | 0,137633 | 12,9267 | 0,0335 |
R = 94,4186
Sg75 WHT
Effects analysis
Cat. Factors:
C=Thickness
Quant. Factors:
A=Speed
B=layer height
Source | Sum of Squares | f.d. | Medium Square | F-Ratio | P-Value |
A | 0,0901333 | 1 | 0,0901333 | 515,048 | 0,0019 |
B | 0,000833333 | 1 | 0,000833333 | 4,7619 | 0,1608 |
C | 0,0835167 | 2 | 0,0417583 | 238,619 | 0,0042 |
AB | 0,0012 | 1 | 0,0012 | 6,85714 | 0,1201 |
AC | 0,00921667 | 2 | 0,00460833 | 26,3333 | 0,0366 |
BC | 0,315317 | 2 | 0,157658 | 900,905 | 0,0011 |
R = 99,9301%
G BLK
Effects analysis
Cat. Factors:
C=Thickness
Quant. Factors:
A=Speed
B=layer height
Source | Sum of Squares | f.d. | Medium Square | F-Ratio | P-Value |
B | 8,21708 | 1 | 8,21708 | 154,432 | 0,0011 |
C | 11,9443 | 2 | 5,97216 | 112,241 | 0,0015 |
AB | 0,0352083 | 1 | 0,0352083 | 0,661707 | 0,4755 |
AC | 0,498017 | 2 | 0,249008 | 4,67987 | 0,1196 |
BC | 1,70385 | 2 | 0,851925 | 16,0111 | 0,0251 |
R = 99,2924%
G WHT
Effects analysis
Cat. Factors:
C=Thickness
Quant. Factors:
A=Speed
B=layer height
Source | Sum of Squares | f.d. | Medium Square | F-Ratio | P-Value |
A | 0,0234083 | 1 | 0,0234083 | 0,431159 | 0,5583 |
B | 9,13508 | 1 | 9,13508 | 168,259 | 0,0010 |
C | 17,4425 | 2 | 8,72123 | 160,637 | 0,0009 |
AC | 0,386867 | 2 | 0,193433 | 3,56285 | 0,1613 |
BC | 1,205 | 2 | 0,6025 | 11,0975 | 0,0411 |
R = 99,4256
Flop BLK
Effects analysis
Cat. Factors:
C=Thickness
Quant. Factors:
A=Speed
B=layer height
Source | Sum of Squares | f.d. | Medium Square | F-Ratio | P-Value |
B | 0,074366 | 1 | 0,074366 | 5,47407 | 0,1013 |
C | 0,480822 | 2 | 0,240411 | 17,6966 | 0,0218 |
AB | 0,00553961 | 1 | 0,00553961 | 0,407769 | 0,5685 |
AC | 0,150533 | 2 | 0,0752666 | 5,54036 | 0,0983 |
BC | 0,0384627 | 2 | 0,0192314 | 1,41562 | 0,3690 |
R = 94,8442%
Flop WHT
Effects analysis
Cat. Factors:
C=Thickness
Quant. Factors:
A=Speed
B=layer height
Source | Sum of Squares | f.d. | Medium Square | F-Ratio | P-Value |
B | 0,116901 | 1 | 0,116901 | 12,4054 | 0,0389 |
C | 0,757818 | 2 | 0,378909 | 40,2095 | 0,0068 |
AB | 0,00861754 | 1 | 0,00861754 | 0,914487 | 0,4095 |
AC | 0,134163 | 2 | 0,0670817 | 7,11866 | 0,0726 |
BC | 0,0393828 | 2 | 0,0196914 | 2,08964 | 0,2701 |
R = 97,3948

Reviewer 2 Report
The revised paper has addressed most of the comments and made some improvement but it is not sufficient in my opinion. As mentioned in the reviews, this work is lack of scientific aspect because it simply uses a device to measure commercial products. The results are presented in pieces instead of a systematic manner, which cannot provide a general knowledge base to the audience. For example, Sparkle grade result varies with the orientation (15, 45, 75) and background (w, b). Despite explanations provided by the authors, they are very specific and cannot link to major findings. Also, apparently the observation angle becomes a variable, which is supposed to be an output.
It is important to note that ANOVA (p-value analysis) simply indicates the ability to distinguish the data but not the level of difference. Interaction plots can better see the effect of each variable, but they are only provided for a couple of cases (Fig. 6 and Fig. 7). Further, a 2-level study cannot really "optimize" but only sees the trend. This is an interesting and perhaps an important topic, but for a broad impact, the authors may consider a more systematic way to present the data in a comprehensive and concise manner to show both significant factors and their effects.
Author Response
REV_2
Comments and Suggestions for Authors
The revised paper has addressed most of the comments and made some improvement but it is not sufficient in my opinion. As mentioned in the reviews, this work is lack of scientific aspect because it simply uses a device to measure commercial products. The results are presented in pieces instead of a systematic manner, which cannot provide a general knowledge base to the audience. For example, Sparkle grade result varies with the orientation (15, 45, 75) and background (w, b). Despite explanations provided by the authors, they are very specific and cannot link to major findings. Also, apparently the observation angle becomes a variable, which is supposed to be an output.
It is important to note that ANOVA (p-value analysis) simply indicates the ability to distinguish the data but not the level of difference. Interaction plots can better see the effect of each variable, but they are only provided for a couple of cases (Fig. 6 and Fig. 7). Further, a 2-level study cannot really "optimize" but only sees the trend. This is an interesting and perhaps an important topic, but for a broad impact, the authors may consider a more systematic way to present the data in a comprehensive and concise manner to show both significant factors and their effects.
The scientific aspects of this work were highlighted in the last version, and we provide references, and equations and new plots with the most interesting results, and to support our hypothesis. As we already argue the statistical approach and colour appearance analysis and discussions were significant and improved. We decided to replace the first interactions plots with the ones in the second version in order to help the readers to understand the final effects depending on the measured angles, the different backgrounds and materials. All the results to ensure the maximum textural effects are in Tables 1-2 and the significant effects founded in all the tested samples are in Tables 5 and 7. We attached, as supplementary material, all the statistically interaction plots and tables. However, in our opinion this information should not be added to the paper, because it could confuse and lost the readers for the significant findings in our paper. On the other hand we add a plot (Figure 9) with the reflectance values of the sample in 12 conditions for the same materials, to reinforce one of our main finding’s, the loss of the diffractive effect under work conditions.
Main changes in the new version:
In the new paper version:
“Furthermore, any filament producer can guarantee final textural effects on the produced 3D printer object. The critical factors in FDM printers have been studied to improve the mechanical properties of the 3D object or reduce printing costs [16]–[20]. Vasudevarao et al. [21] proposed an experimental design to determine significant factors and their interactions for optimal surface finish of parts fabricated via Fused Deposition Modelling process. As Rapid Prototyping is moving towards Rapid Manufacturing there is an increasing stress on obtaining good quality parts. Quality of a prototype includes the surface quality, the mechanical strength and dimensional accuracy among other things. Surface Finish is critical not only for better functionality and look, but also for cost reduction in terms of reduced post-processing of parts (this includes sanding, filing etc.) and overall prototyping time reduction also. However, the effect of 3D printing parameters on the final visual appearance has not yet been analysed. The aim of this work was to control the metallic and interference effect on 3D-printed objects by changing certain printing parameters, such as filament material, sample thickness, printing speed, etc.
2. Materials and Methods
We used the free software platform Blender for the 3D objects designs. To build the 3D samples, an Ultimaker 2+ 3D was used. This printer machine is capable of working with different nozzles sizes: 0.25, 0.4, 0.6 and 0.8 mm. This machine works with a 2.85 mm filament diameter. However, changing printing parameters is possible to work with different filament diameters. In addition we used CURA software, designed by Ultimaker (Umetani, Bickel, Matusik, 2015), to change the sample and 3D printing parameters.
For this study, work was done with two commercial filaments from different polymers and with distinct properties. PLA grey and metallic filaments were used (2.85 mm in diameter), obtained from Utimaker (UM9015 PLA Silver Metallic RAL 9006), along with ABS grey with 3% diffractive pigments (MAXITHEN® SAN702767/10 Metallic Silver), and a diameter of 1.75 mm, which came from Schlenk Metallic Pigments GmbH (Figure 1). The last filament was not available in the market before our test. Schlenk Company was interested to know the diffractive pigments behaviour under work conditions. This work is going to provide critical information to the filament makers, as the final effects limits to this new filaments.
Results and discussion_ line 296:
The most significant factor to achieve high flop values and Sg75_b was layer height (0.1mm). The lower the layer height, the higher the flop value on the black or white background (Table 7). This may be due to the diffractive pigment orientations in thinner layers. However, this parameter did not suffice to maintain the diffractive effect, and only in sparkle was it possible to see different colours change the direct or diffuse illumination. The textural and final metallic effects were the same in both samples PLA and ABS, as seen in Figure 8. The visual appearance must change significantly with the orientation angle of measurement, mostly in the ABS filament with diffractive pigments. However, the “rainbow” effect disappears in working conditions. The reflectance spectrum values in VIS range change mainly in parallel as the PLA samples with metallic pigments, and is not possible to observe significant differences in the band positions or the spectrum shapes with the change of the orientation measurement [23]. That means mainly changes in L*ab value due to the orientation measurement changes, but no difference colours can be observed due to the diffractive pigments, as its represented in the example (Figure 9). The effect that was firstly observed in the filament was lost in the melting process of 3D printing. The different orientations of layers during the printing process did not allows the diffraction effect to be controlled and maintained as in the raw material.
Figure 9. VIS reflectance spectrum values from PLA and ABS sample in 12 DoE conditions
4. Conclusions
The aim of this work was to find significant differences in the textural effect of 3D-printed objects using different filament materials with metallic and diffractive pigments, and to discover the optimal conditions that achieve the maximum sparkle, flop and graininess effects. In addition, a filament producer company Schlenk, was interested to know the diffractive pigments behaviour under work conditions for the filament prototype that they provide for this work. As pigments manufacturers they are focused in innovation to develop coatings, and materials interesting for advanced applications as automotive sector. With the multivariate factorial DoE, it was possible to define the most significant printing factors for both filaments ABS prototype and PLA with metallic pigments, improving the gonio-appearance of the 3D-printed objects. The best conditions to increase the metallic effects of the PLA objects at the 15º observation angle are achieved at a low layer height (0.1 mm) and low sample thickness. In addition, it is recommended to use low printing speed to allow the effect pigments particles orientation. The graininess effect is stronger when layer height is increased (0.2 mm), so working at high layer levels is recommended if increasing the diffuse coarseness surfaces is important. However, this measurement could be affected for the surface roughness of FDM samples, and this should be taken into account. For this reason, sparkle and flop parameters are better to analyse the printing factor effects with FDM 3D objects. To increase the flop parameter (on a black or a white background), increasing sample thickness is recommended.
SUPPLEMENTARY MATERIAL: ANOVA TEST AND INTERACTION PLOTS
ABS
Sg45 BLK
Effects analysis
Cat. Factors:
C=Thickness
Quant. Factors:
A=Speed
B=layer height
Source | Sum of Squares | f.d. | Medium Square | F-Ratio | P-Value |
A | 0,0420083 | 1 | 0,0420083 | 387,769 | 0,0026 |
B | 0,308291 | 1 | 0,308291 | 2845,76 | 0,0004 |
C | 0,320095 | 2 | 0,160048 | 1477,36 | 0,0007 |
AB | 0,725208 | 1 | 0,725208 | 6694,23 | 0,0001 |
AC | 0,147117 | 2 | 0,0735583 | 679,0 | 0,0015 |
BC | 0,03065 | 2 | 0,015325 | 141,462 | 0,0070 |
R = 99,9984%
Sg45 WHT
Effects analysis
Cat. Factors:
C=Thickness
Quant. Factors:
A=Speed
B=layer height
Source | Sum of Squares | f.d. | Medium Square | F-Ratio | P-Value |
A | 0,195075 | 1 | 0,195075 | 1,58141 | 0,2641 |
B | 6,64541 | 1 | 6,64541 | 53,8722 | 0,0007 |
C | 1,46622 | 2 | 0,733108 | 5,94308 | 0,0477 |
BC | 0,451017 | 2 | 0,225508 | 1,82812 | 0,2536 |
R = 93,4207%
Sg15 BLK
Effects analysis
Cat. Factors:
C=Thickness
Quant. Factors:
A=Speed
B=layer height
Source | Sum of Squares | f.d. | Medium Square | F-Ratio | P-Value |
A | 0,6348 | 1 | 0,6348 | 3,75592 | 0,1104 |
B | 14,564 | 1 | 14,564 | 86,1709 | 0,0002 |
C | 1,15962 | 2 | 0,579812 | 3,43057 | 0,1154 |
AC | 1,15755 | 2 | 0,578775 | 3,42443 | 0,1157 |
R= 95,0884%
Sg15 WHT
Effects analysis
Cat. Factors:
C=Thickness
Quant. Factors:
A=Speed
B=layer height
Source | Sum of Squares | f.d. | Medium Square | F-Ratio | P-Value |
A | 0,567675 | 1 | 0,567675 | 4,13632 | 0,1349 |
B | 10,811 | 1 | 10,811 | 78,7735 | 0,0030 |
C | 0,957596 | 2 | 0,478798 | 3,48872 | 0,1649 |
AC | 1,02305 | 2 | 0,511525 | 3,72718 | 0,1537 |
BC | 0,266217 | 2 | 0,133108 | 0,969883 | 0,4733 |
R = 97,173%
Sg75 BLK
Effects analysis
Cat. Factors:
C=Thickness
Quant. Factors:
A=Speed
B=layer height
Source | Sum of Squares | f.d. | Medium Square | F-Ratio | P-Value |
B | 1,21157 | 1 | 1,21157 | 5,83945 | 0,0604 |
C | 0,802248 | 2 | 0,401124 | 1,93331 | 0,2388 |
AB | 1,1907 | 1 | 1,1907 | 5,73887 | 0,0620 |
AC | 1,03752 | 2 | 0,518758 | 2,50028 | 0,1768 |
R = 82,4463%
Sg75 WHT
Effects analysis
Cat. Factors:
C=Thickness
Quant. Factors:
A=Speed
B=layer height
Source | Sum of Squares | f.d. | Medium Square | F-Ratio | P-Value |
A | 0,00213333 | 1 | 0,00213333 | 0,00505399 | 0,9498 |
B | 1,25624 | 1 | 1,25624 | 2,9761 | 0,2266 |
C | 0,693334 | 2 | 0,346667 | 0,821275 | 0,5491 |
AB | 1,25453 | 1 | 1,25453 | 2,97206 | 0,2269 |
AC | 0,975317 | 2 | 0,487658 | 1,15529 | 0,4640 |
BC | 0,624817 | 2 | 0,312408 | 0,740114 | 0,5747 |
R = 85,8947%
G BLK
Effects analysis
Cat. Factors:
C=Thickness
Quant. Factors:
A=Speed
B=layer height
Source | Sum of Squares | f.d. | Medium Square | F-Ratio | P-Value |
A | 0,00653333 | 1 | 0,00653333 | 0,0619127 | 0,8267 |
B | 0,185149 | 1 | 0,185149 | 1,75455 | 0,3164 |
C | 0,352667 | 2 | 0,176333 | 1,67101 | 0,3744 |
AB | 0,2352 | 1 | 0,2352 | 2,22886 | 0,2740 |
AC | 0,303517 | 2 | 0,151758 | 1,43813 | 0,4102 |
BC | 0,0193167 | 2 | 0,00965833 | 0,0915265 | 0,9161 |
R = 85,0633%
G WHT
Effects analysis
Cat. Factors:
C=Thickness
Quant. Factors:
A=Speed
B=layer height
Source | Sum of Squares | f.d. | Medium Square | F-Ratio | P-Value |
A | 0,0147 | 1 | 0,0147 | 0,153712 | 0,7328 |
B | 0,118021 | 1 | 0,118021 | 1,2341 | 0,3823 |
C | 0,355874 | 2 | 0,177937 | 1,86062 | 0,3496 |
AB | 0,154133 | 1 | 0,154133 | 1,61171 | 0,3320 |
AC | 0,3318 | 2 | 0,1659 | 1,73475 | 0,3657 |
BC | 0,0866667 | 2 | 0,0433333 | 0,45312 | 0,6882 |
R = 92,7219%
Flop BLK
Effects analysis
Cat. Factors:
C=Thickness
Quant. Factors:
A=Speed
B=layer height
Source | Sum of Squares | f.d. | Medium Square | F-Ratio | P-Value |
A | 0,0444929 | 1 | 0,0444929 | 0,404272 | 0,5701 |
B | 5,40393 | 1 | 5,40393 | 49,1013 | 0,0060 |
C | 0,123023 | 2 | 0,0615116 | 0,558908 | 0,6218 |
AC | 0,110493 | 2 | 0,0552464 | 0,501981 | 0,6486 |
BC | 0,0396714 | 2 | 0,0198357 | 0,180231 | 0,8435 |
R = 94,6489%
Flop WHT
Effects analysis
Cat. Factors:
C=Thickness
Quant. Factors:
A=Speed
B=layer height
Source | Sum of Squares | f.d. | Medium Square | F-Ratio | P-Value |
A | 0,155358 | 1 | 0,155358 | 1,96926 | 0,2195 |
B | 4,56695 | 1 | 4,56695 | 57,8888 | 0,0006 |
C | 0,180596 | 2 | 0,0902981 | 1,14458 | 0,3897 |
AC | 0,199523 | 2 | 0,0997616 | 1,26454 | 0,3594 |
R= 93,295
PLA
Sg45 BLK
Effects analysis
Cat. Factors:
C=Thickness
Quant. Factors:
A=Speed
B=layer height
Source | Sum of Squares | f.d. | Medium Square | F-Ratio | P-Value |
A | 0,00100833 | 1 | 0,00100833 | 0,0425906 | 0,8497 |
B | 0,378075 | 1 | 0,378075 | 15,9694 | 0,0281 |
C | 0,18035 | 2 | 0,090175 | 3,80887 | 0,1502 |
AC | 0,0367167 | 2 | 0,0183583 | 0,775431 | 0,5352 |
BC | 0,06845 | 2 | 0,034225 | 1,44562 | 0,3634 |
R = 90,3449%
Sg45 WHT
Effects analysis
Cat. Factors:
C=Thickness
Quant. Factors:
A=Speed
B=layer height
Source | Sum of Squares | f.d. | Medium Square | F-Ratio | P-Value |
B | 0,0660083 | 1 | 0,0660083 | 9,64799 | 0,0530 |
C | 0,05345 | 2 | 0,026725 | 3,90621 | 0,1461 |
AB | 0,039675 | 1 | 0,039675 | 5,79903 | 0,0952 |
AC | 0,11255 | 2 | 0,056275 | 8,22533 | 0,0606 |
BC | 0,164017 | 2 | 0,0820083 | 11,9866 | 0,0371 |
R = 95,5011%
Sg15 BLK
Effects analysis
Cat. Factors:
C=Thickness
Quant. Factors:
A=Speed
B=layer height
Source | Sum of Squares | f.d. | Medium Square | F-Ratio | P-Value |
A | 0,00240833 | 1 | 0,00240833 | 0,0280338 | 0,8824 |
B | 0,143008 | 1 | 0,143008 | 1,66466 | 0,3260 |
C | 0,24605 | 2 | 0,123025 | 1,43205 | 0,4112 |
AB | 0,0602083 | 1 | 0,0602083 | 0,700844 | 0,4906 |
AC | 0,0917167 | 2 | 0,0458583 | 0,533805 | 0,6520 |
BC | 0,0862167 | 2 | 0,0431083 | 0,501795 | 0,6659 |
R = 78,5611
Sg15 WHT
Effects analysis
Cat. Factors:
C=Thickness
Quant. Factors:
A=Speed
B=layer height
Source | Sum of Squares | f.d. | Medium Square | F-Ratio | P-Value |
B | 0,226875 | 1 | 0,226875 | 10,5851 | 0,0313 |
C | 0,10365 | 2 | 0,051825 | 2,41796 | 0,2049 |
AC | 0,0877167 | 2 | 0,0438583 | 2,04627 | 0,2443 |
BC | 0,00945 | 2 | 0,004725 | 0,220451 | 0,8113 |
R = 83,3017%
Sg75 BLK
Effects analysis
Cat. Factors:
C=Thickness
Quant. Factors:
A=Speed
B=layer height
Source | Sum of Squares | f.d. | Medium Square | F-Ratio | P-Value |
B | 0,00300833 | 1 | 0,00300833 | 0,282546 | 0,6319 |
C | 0,211667 | 2 | 0,105833 | 9,93999 | 0,0475 |
AB | 0,00300833 | 1 | 0,00300833 | 0,282546 | 0,6319 |
AC | 0,0474 | 2 | 0,0237 | 2,22593 | 0,2554 |
BC | 0,275267 | 2 | 0,137633 | 12,9267 | 0,0335 |
R = 94,4186
Sg75 WHT
Effects analysis
Cat. Factors:
C=Thickness
Quant. Factors:
A=Speed
B=layer height
Source | Sum of Squares | f.d. | Medium Square | F-Ratio | P-Value |
A | 0,0901333 | 1 | 0,0901333 | 515,048 | 0,0019 |
B | 0,000833333 | 1 | 0,000833333 | 4,7619 | 0,1608 |
C | 0,0835167 | 2 | 0,0417583 | 238,619 | 0,0042 |
AB | 0,0012 | 1 | 0,0012 | 6,85714 | 0,1201 |
AC | 0,00921667 | 2 | 0,00460833 | 26,3333 | 0,0366 |
BC | 0,315317 | 2 | 0,157658 | 900,905 | 0,0011 |
R = 99,9301%
G BLK
Effects analysis
Cat. Factors:
C=Thickness
Quant. Factors:
A=Speed
B=layer height
Source | Sum of Squares | f.d. | Medium Square | F-Ratio | P-Value |
B | 8,21708 | 1 | 8,21708 | 154,432 | 0,0011 |
C | 11,9443 | 2 | 5,97216 | 112,241 | 0,0015 |
AB | 0,0352083 | 1 | 0,0352083 | 0,661707 | 0,4755 |
AC | 0,498017 | 2 | 0,249008 | 4,67987 | 0,1196 |
BC | 1,70385 | 2 | 0,851925 | 16,0111 | 0,0251 |
R = 99,2924%
G WHT
Effects analysis
Cat. Factors:
C=Thickness
Quant. Factors:
A=Speed
B=layer height
Source | Sum of Squares | f.d. | Medium Square | F-Ratio | P-Value |
A | 0,0234083 | 1 | 0,0234083 | 0,431159 | 0,5583 |
B | 9,13508 | 1 | 9,13508 | 168,259 | 0,0010 |
C | 17,4425 | 2 | 8,72123 | 160,637 | 0,0009 |
AC | 0,386867 | 2 | 0,193433 | 3,56285 | 0,1613 |
BC | 1,205 | 2 | 0,6025 | 11,0975 | 0,0411 |
R = 99,4256
Flop BLK
Effects analysis
Cat. Factors:
C=Thickness
Quant. Factors:
A=Speed
B=layer height
Source | Sum of Squares | f.d. | Medium Square | F-Ratio | P-Value |
B | 0,074366 | 1 | 0,074366 | 5,47407 | 0,1013 |
C | 0,480822 | 2 | 0,240411 | 17,6966 | 0,0218 |
AB | 0,00553961 | 1 | 0,00553961 | 0,407769 | 0,5685 |
AC | 0,150533 | 2 | 0,0752666 | 5,54036 | 0,0983 |
BC | 0,0384627 | 2 | 0,0192314 | 1,41562 | 0,3690 |
R = 94,8442%
Flop WHT
Effects analysis
Cat. Factors:
C=Thickness
Quant. Factors:
A=Speed
B=layer height
Source | Sum of Squares | f.d. | Medium Square | F-Ratio | P-Value |
B | 0,116901 | 1 | 0,116901 | 12,4054 | 0,0389 |
C | 0,757818 | 2 | 0,378909 | 40,2095 | 0,0068 |
AB | 0,00861754 | 1 | 0,00861754 | 0,914487 | 0,4095 |
AC | 0,134163 | 2 | 0,0670817 | 7,11866 | 0,0726 |
BC | 0,0393828 | 2 | 0,0196914 | 2,08964 | 0,2701 |
R = 97,3948
